# Cost–Utility Analysis of PCSK9 Inhibitors and Quality of Life: A Two-Year Multicenter Non-Randomized Study

**DOI:** 10.3390/diseases12100244

**Published:** 2024-10-05

**Authors:** José Seijas-Amigo, Maria José Mauriz-Montero, Pedro Suarez-Artime, Mónica Gayoso-Rey, Francisco Reyes-Santías, Ana Estany-Gestal, Antonia Casas-Martínez, Lara González-Freire, Ana Rodriguez-Vazquez, Natalia Pérez-Rodriguez, Laura Villaverde-Piñeiro, Concepción Castro-Rubinos, Esther Espino-Paisán, Octavio Cordova-Arevalo, Diego Rodriguez-Penas, Begoña Cardeso-Paredes, Marta Ribeiro-Ferreiro, Moisés Rodríguez-Mañero, Alberto Cordero, José R. González-Juanatey

**Affiliations:** 1Cardiology Department, Complejo Hospitalario Universidad de Santiago de Compostela, Santiago de Compostela, 15706 Santiago de Compostela, Spain; diego.rodriguez.penas@sergas.es (D.R.-P.); begona.cardeso.paredes@sergas.es (B.C.-P.); marta.ribeiro.ferreiro@sergas.es (M.R.-F.); moirmanero@gmail.com (M.R.-M.); jose.ramon.gonzalez.juanatey@sergas.es (J.R.G.-J.); 2Fundación Instituto de Investigación Sanitaria de Santiago de Compostela (FIDIS), 15706 Santiago de Compostela, Spain; ana.estany.gestal@sergas.es; 3Centro de Investigación Biomédica en Red de Enfermedades Cardiovasculares (CIBERCV), 28029 Madrid, Spain; acorderofort@gmail.com; 4Pharmacy Department, Complejo Hospitalario Universitario A Coruña, 15006 A Coruña, Spain; ma.jose.mauriz.montero@sergas.es; 5Pharmacy Department, Complejo Hospitalario Universidad de Santiago de Compostela, 15782 Santiago de Compostela, Spain; pedro.suarez.artime@sergas.es; 6Pharmacy Department, Complejo Hospitalario Universitario de Vigo, 36312 Vigo, Spain; monica.gayoso.rey@sergas.es; 7Management Department, Complejo Hospitalario Universidad de Santiago de Compostela, 15782 Santiago de Compostela, Spain; francisco.reyes.santias@sergas.es; 8Pharmacy Department, Complejo Hospitalario Universitario de Ferrol, 15405 Ferrol, Spain; antonia.casas.martinez@sergas.es; 9Pharmacy Department, Complejo Hospitalario Universitario de Pontevedra, 36472 Pontevedra, Spain; lara.gonzalez.freire@sergas.es; 10Pharmacy Department, Complejo Hospitalario Universitario de Ourense, 32005 Ourense, Spain; ana.rodriguez.vazquez2@sergas.es; 11Pharmacy Department, Complejo Hospitalario Universitario de Lugo, 27003 Lugo, Spain; natalia.perez.rodriguez@sergas.es; 12Pharmacy Department, Hospital Comarcal de Monforte, 27400 Monforte de Lemos, Spain; laura.villaverde.pineiro@sergas.es; 13Pharmacy Department, Hospital Público da Mariña, 27880 Burela, Spain; concepcion.castro.rubinos@sergas.es; 14Pharmacy Department, Hospital do Barbanza, 15993 A Coruña, Spain; esther.espino.paisan@sergas.es; 15PEMEX, Universidade de Vigo, 36310 Vigo, Spain; octavio.oficina@gmail.com; 16Cardiology Department, Cardiology Department Hospital IMED Elche, 03203 Elche, Spain; 17Unidad de Investigación en Cardiología, Fundación para el Fomento de la Investigación Sanitaria y Biomédica de la Comunitat Valenciana (FISABIO), 46020 Valencia, Spain

**Keywords:** PCSK9 inhibitors, cost–utility analysis, quality of life, real-world study, cardiovascular disease

## Abstract

The primary objective of this study was to conduct a cost–utility analysis of proprotein convertase subtilisin/kexin type 9 (PCSK9) inhibitors in real-world, comparing their use with standard care for managing cardiovascular disease. A multicenter prospective study was conducted across 12 Spanish hospitals from May 2020 to April 2022, involving 158 patients with hypercholesterolemia or atherosclerotic cardiovascular disease. This study assessed health-related quality of life (QoL) using the EQ-5D-3L questionnaire. The cost–utility analysis evaluated the economic impact of PCSK9 inhibitors when used with standard care compared to standard care alone, calculating the incremental cost–effectiveness ratio (ICER). This study included 158 patients with an average age of 61 years, male (66.5%). For patients initiating PCSK9 inhibitors, the treatment cost was EUR 13,633.39, while standard therapy cost EUR 3638.25 over two years. QoL for PCSK9 inhibitors stood at 1.6489 over two years, compared to 1.4548 for standard therapy. The results revealed favorable cost–utility outcomes, with an ICER of EUR 51,427.72. Significant improvements were observed in the domains of mobility, self-care, daily activities, pain/discomfort, and anxiety/depression (*p* < 0.001). This study presents the first real-world cost–utility analysis of PCSK9 inhibitors, supporting their economic rationale and highlighting their benefits in clinical practice. Healthcare decision-makers can use these results to inform their decisions and reimbursement policies concerning PCSK9 inhibitors. Trial Registration clinicaltrials.gov Identifier: NCT04319081.

## 1. Introduction

Cardiovascular diseases (CVDs) remain a leading cause of morbidity and mortality worldwide, imposing a substantial economic burden on healthcare systems [1]. Elevated low-density lipoprotein cholesterol (LDL-C) levels have been identified as a major risk factor for CVD development and progression. In recent years, the advent of novel therapeutic agents, such as proprotein convertase subtilisin/kexin type 9 (PCSK9) inhibitors, has provided promising avenues for effectively managing dyslipidemia [2].

PCSK9 inhibitors, such as evolocumab and alirocumab, have demonstrated remarkable LDL-C-lowering efficacy in randomized controlled trials (RCTs), leading to their approval and integration into clinical practice guidelines [3,4]. However, the economic implications of these new therapies, especially in real-world settings, have garnered significant attention. Real-world studies provide crucial insights into the cost effectiveness and value of interventions when implemented in routine clinical practice [5].

The economic burden of CVD in the European Union is substantial, costing EUR 282 billion annually. Health- and long-term care account for EUR 155 billion, or 55% of this total, equating to 11% of EU health expenditure. Productivity losses due to mortality and morbidity contribute EUR 48 billion, representing 17% of the total cost. Informal care, essential yet often overlooked, accounts for EUR 79 billion, or 28%. The per capita cost of CVD across the EU averages EUR 630, with notable variations between countries, ranging from EUR 381 in Cyprus to EUR 903 in Germany. The significant economic impact of CVD underscores the urgent need for efficient allocation of resources towards prevention, treatment, and management within the healthcare system to mitigate its burden on the economy and improve health outcomes across the European Union [6].

In patients with familial hypercholesterolemia (HF) or those requiring secondary prevention for dyslipidemia, where statins fail to adequately control cholesterol due to intolerance or refractoriness, the European Medicines Agency has sanctioned the use of alirocumab and evolocumab. Despite their commercialization at significantly higher prices compared to other cholesterol-lowering medications, and the absence of comprehensive morbidity and mortality research, these drugs represent a step towards optimizing treatment efficiency within constrained resource settings [7].

Real-world evidence becomes crucial in this context, offering insights beyond the controlled environments of RCTs. It highlights the practical value of interventions, including their impact on health-related quality of life (QoL) and their economic viability, through cost–utility analysis (CUA). Unlike traditional cost–effectiveness analysis (CEA), which focuses solely on the economic impact, CUA emphasizes the quality and quantity of life gained, using measures like the EuroQol-5 Dimension (EQ-5D-3L) questionnaire to assess changes in QoL. This approach provides a more holistic view of the value PCSK9 inhibitors add, balancing their cost against the health benefits they offer.

Currently, there is a lack of cost–utility data derived from real-world studies for PCSK9 inhibitors, as quality of life and cost–utility variables are typically assessed primarily within the context of clinical trials. However, comprehending the economic impact and utility of these inhibitors in everyday clinical practice is vital for making well-informed choices [8].

The present study aims to investigate the cost–utility and quality of life outcomes of PCSK9 inhibitors using prospectively collected data from 158 patients over a 24-month follow-up period. The primary objective of this study is to assess the cost–utility of PCSK9 inhibitors, while the secondary objective was to evaluate changes in quality of life measured through the different domains of the EQ-5D-3L questionnaire. By examining these comprehensive measures, this study aims to provide valuable insights into the economic impact and utility of PCSK9 inhibitors in real-world clinical practice, building upon the findings from the MEMOGAL study (ClinicalTrials.gov Identifier: NCT04319081).

## 2. Results

### 2.1. Patient Disposition

From 25 May 2020 (first patient) to 6 April 2022 (last patient included), a total of 158 patients were enrolled for the study followed for a median of 99 weeks. All participants successfully completed the final EQ-5D-3L questionnaire, except for those who were deceased (n = 2), for whom data were collected during previous visits. (Figure 1). Additionally, all 158 patients were retrospectively used as a control group before the initiation of PCSK9 inhibitors treatment while they were on standard therapy (statins and/or ezetimibe).

### 2.2. Baseline Characteristics and Concomitant Medication

Table 1 presents the baseline characteristics of the study population. The patients had a mean (SD) age of 61 (10) years, with 66.5% being male. The mean (SD) body weight and BMI were 81 (16) kg and 29 (5) kg/m^2^, respectively. Among the participants, 85% had cardiovascular ischemic disease (CVD), 25% had FH, 55% had hypertension, 22% had type 2 diabetes (T2D), and 17% had heart failure. Comorbidities included a family history of dementia in 20% of patients, 11% were smokers, and 72% adhered to a diet.

Out of the 158 patients, 75 were receiving evolocumab 140 mg every 2 weeks (47.46%), 65 were on alirocumab 150 mg every 2 weeks (41.14%), and 18 were on alirocumab 300 mg every 2 weeks (11.40%). As for additional lipid-lowering therapy, 33.5% were taking rosuvastatin, 18.4% were taking atorvastatin, 3.2% were taking pitavastatin, 1.2% were taking other statins, and 58.6% were taking ezetimibe. Notably, 43.7% of the sample was not taking any statin due to statin intolerance.

## 3. Outcomes

### 3.1. Primary Endpoint: Cost–Utility

We conducted an analysis to calculate the gained QoL and associated costs for both treatments, PCSK9 inhibitors and standard therapy.

The events and costs are shown in Table 2.

Results (Table 3):

ICER: (Total unit cost PCSK9 inhibitors—Total unit cost standard therapy)/(QoL PCSK9 inhibitors—QoL standard therapy) = EUR 51,427.72

### 3.2. Secondary Endpoint: Changes in QoL

At the end of the study, there were no missing data for any patients (0%) during the follow-up period. A total of 158 subjects were included in the analysis. The percentage of changes in the EQ-5D-3L domains and the total VAS score were calculated (see Table 4).

In the baseline assessment of the mobility domain, the distribution of symptoms among patients was as follows: 77.2% reported no symptoms, 21.5% had moderate symptoms, and 1.3% experienced severe symptoms. However, at follow-up, the distribution shifted significantly, with 87.7% of patients reporting no symptoms, 12.3% experiencing moderate symptoms, and no patients reporting severe symptoms. This observed improvement in mobility outcomes was statistically significant (*p* < 0.001).

In the self-care domain, at baseline, the distribution of patients’ self-care abilities was as follows: 90.6% reported no difficulties, 9.4% had moderate difficulties, and 1.3% experienced severe difficulties. However, at follow-up, there was a significant improvement in self-care outcomes, with 97% of patients reporting no difficulties, 3% experiencing moderate difficulties, and none of the patients reporting severe difficulties.

In the daily activities domain, at baseline, the distribution of patients’ abilities to perform daily activities was as follows: 86.6% reported no difficulties, 12.8% had moderate difficulties, and 0.6% experienced severe difficulties. However, at follow-up, there was a significant improvement in daily activities outcomes, with 94.1% of patients reporting no difficulties, 5.9% experiencing moderate difficulties, and none of the patients reporting severe difficulties.

In the baseline assessment of the pain or discomfort domain, the distribution of symptoms among patients was as follows: 48.6% reported no pain or discomfort, 45.3% had moderate pain or discomfort, and 6.1% experienced severe pain or discomfort. However, at follow-up, the distribution shifted significantly, with 68.4% of patients reporting no pain or discomfort, 31.6% experiencing moderate pain or discomfort, and no patients reporting severe pain or discomfort. This observed improvement in pain or discomfort outcomes was statistically significant (*p* < 0.001).

Finally, regarding the baseline assessment of anxiety or depression domain, the distribution of symptoms among patients was as follows: 64.4% reported no anxiety or depression, 28.2% had moderate levels of anxiety or depression, and 7.4% experienced severe levels. However, at follow-up, the distribution shifted significantly, with 80.6% of patients reporting no anxiety or depression, 14.6% experiencing moderate levels, and 4.9% reporting severe levels. This observed improvement in anxiety or depression outcomes was statistically significant (*p* < 0.001).

The mean change in VAS score from baseline was 67.04 ( ± 20.069) (95% CI 66.62-72.67) to follow-up, with a value of 69.64 ( ± 18.683) (95% CI 69.64-18.683) (*p* = 0.086). This represents an increase of 2.6 points, although the difference was not statistically significant (see Figure 2).

Furthermore, the values of the Spanish population norms (EQ-5D-3L) are included in Table 5, which provides a detailed breakdown by age and gender for the Spanish population norms (EQ-5D-3L), including mean EQ-VAS scores and the percentage of patients reporting problems across dimensions.

## 4. Discussion

The analyses conducted in this observational and prospective study, which included real-world patients treated with PCSK9 inhibitors, provide valuable insights into the cost–utility of PCSK9 inhibitors in this population. The results of the study demonstrate that the use of PCSK9 inhibitors was associated with favorable cost–utility outcomes, indicating its potential economic value in real-world clinical practice. Additionally, we also assessed changes in quality of life using the EQ-5D-3L questionnaire among 158 real-world patients followed from the initiation of treatment with PCSK9 inhibitors for a duration of 2 years. The results obtained indicate an improvement in all domains: mobility, self-care, activities, pain, and anxiety and/or depression. Furthermore, there was an improvement observed in the overall VAS EQ-5D-3L score.

Among the 158 patients included in this study, the groups taking alirocumab and evolocumab were almost equally distributed, and 158 patients completed the EQ-5D-3L questionnaire at follow-up (100%) (Figure 1). The main reduction in LDL-c levels was 55.6% (from 145.18 mg/dl to 62.11 mg/dL), which closely aligns with the results reported in the pivotal RCT [3,4]. Similarly, it is worth highlighting the subsequent role of small interfering RNA (siRNA) targeting PCSK9, such as Inclisiran, which has shown LDL-C reductions very similar (around 50%) to those reported in this study [10]. Major Adverse Cardiovascular Events (MACEs) were observed in 10 patients following the initiation of PCSK9 inhibitors during a 2-year follow-up period. As depicted in Table 2, we also retrospectively recorded events related to standard therapy from the electronic medical record system (IANUS), which covers all hospitals and primary care centers in this area. All direct and indirect costs were obtained from the Ministry of Health, and the average costs of Diagnosis-Related Groups (DRG) [11] were utilized, with appropriate discounts and updates applied.

In relation to the primary outcome, the calculated incremental cost–effectiveness ratio (ICER) was EUR 51,427.72. This value suggests that the use of PCSK9 inhibitors demonstrates favorable cost–utility across various scenarios and when compared to different comparative references. The ICER serves as an important indicator of the efficiency and economic value of an intervention, and the calculated value supports the cost effectiveness of utilizing PCSK9 inhibitors in the context of this study. These findings highlight the potential benefits and value of incorporating PCSK9 inhibitors into clinical practice, considering their impact on both cost and utility in comparison to alternative therapies.

One of the economic scientific comparators, A. Laupacis et al. [12], indicates a threshold of EUR 51,278.21 (USD 55,000), which, when updated to January 2023, would be EUR 94,076.4. This suggests that the intervention clearly demonstrates cost–utility. When referring to Plans P. et al. as a comparative reference [13], the results also indicate cost–utility, as the economic threshold for this comparator is EUR 46,616,562 (USD 50,000). If we update it to January 2023, the ICER limit would be EUR 68,420. Considering the De Cock and González-Juanatey study [14], which indicates a threshold range between EUR 12,000 and EUR 45,000, even with the current updated value of EUR 52,965 for January 2023, the study still demonstrates cost–utility. Finally, the scenario suggested by the CHOICE project of the World Health Organization (WHO) [15] considers a drug to be cost-effective if its cost–utility is between 1 and 3 times the Gross Domestic Product (GDP) per capita. In Spain, the GDP per capita in 2022 was EUR 27,820, which aligns with the results of this study.

Regarding changes in EQ-5D-3L, it is evident that significant improvements were observed in all domains of the EQ-5D-3L questionnaire. Furthermore, when comparing the baseline and follow-up scores, the domains with the most pronounced improvements were pain/discomfort and anxiety/depression. It is important to note that although improvements were observed across all domains, the pain/discomfort and anxiety/depression domains showed the most remarkable changes. These findings suggest that the use of PCSK9 inhibitors may have a particularly beneficial effect on reducing pain, discomfort, anxiety, and depression among the study population. Our findings align with the direction observed in clinical trials regarding the positive effects of PCSK9 inhibitors on quality of life. For instance, the FOURIER and ODYSSEY trials [3,4] reported improvements in health-related quality of life measures among patients treated with PCSK9 inhibitors compared to standard therapy.

When comparing patient-reported issues from our study to those reported by the Spanish population through various European questionnaires (across 20 countries with N = 163,838; Janssen et al. study [9]), the analysis suggests that the percentage of patients reporting problems in the standard treatment group was similar for daily activities (12% vs. 14%) but was slightly higher in the domains of mobility and self-care (22% vs. 16% and 10% vs. 6%, respectively). However, in the domains of pain and anxiety, the percentage in the standard treatment group compared to the general population was significantly higher (51% vs. 27% and 35% vs. 10%), indicating that cardiovascular disease patients tend to report more issues with pain and anxiety than the general population. Conversely, for patients treated with PCSK9 inhibitors, the percentage of reported problems in mobility, self-care, and daily activities was significantly reduced, even falling below the general population levels (12% vs. 16%; 3% vs. 6%; and 6% vs. 14%, respectively). Likewise, in the domains of pain and anxiety, the figures dropped to nearly match those of the general population (31% vs. 27% and 20% vs. 10%).

Regarding the mean EQ VAS scores, where the general population has an average score of 72, patients on standard treatment scored 67, while those treated with PCSK9 inhibitors scored between 69 and 70, showing a similar positive trend.

This comparison underscores the effectiveness of the interventions in our study at enhancing health-related quality of life among cardiac patients, moving them towards the health status of the general population. It also highlights the significance of targeted healthcare interventions in managing chronic conditions such as CVD, aiming to achieve outcomes that mirror the health status of the broader population.

To the best of our knowledge, this is the first real-world cost–utility study to date. Most of the existing studies have focused on cost effectiveness rather than cost–utility. For example, Samad Azari et al. [16] conducted a cost–effectiveness systematic review comparing PCSK9 inhibitors with standard therapy which does not directly align with our cost–utility analysis. It is important to note that our study results may differ from those of clinical trials. One possible explanation is that patients in our real-world study, treated with inhibitors, may have experienced fewer events compared to patients in clinical trials. This difference could be attributed to various factors, including differences in patient characteristics or comorbidities, and treatment adherence. Additionally, the duration of follow-up in our study may have been shorter than that of clinical trials, which could influence the occurrence of events. On the other hand, several published studies [11,17,18,19] were in line with our study, and they explored cost effectiveness but none of them studied cost–utility. Based on these considerations, we believe that this study provides a novel economic analysis on the real-world impact of PCSK9 inhibitors. However, further comparative studies in this regard will be necessary to strengthen the evidence base. These additional studies can contribute to a more comprehensive understanding of the cost–utility profile of PCSK9 inhibitors and help guide decision-making in clinical practice and healthcare policy.

Further research and longer-term follow-up are needed to fully understand the comparative outcomes and economic implications of PCSK9 inhibitors in real-world settings.

## 5. Limitations

Despite the valuable insights provided by our study, certain limitations should be acknowledged. As an observational study, it is subject to inherent biases and confounding factors, despite efforts to minimize these through robust data collection and statistical adjustments. The reliance on retrospective data collection for some variables, such as events and costs associated with standard therapy, could impact the results due to variability in the completeness and accuracy of medical records. Additionally, the control and treatment groups consisting of the same patients present a limitation, although this was mitigated by the data collection spanning two years prior to the initiation of PCSK9 inhibitors treatment. The relatively short follow-up period of two years limits the understanding of long-term cost–utility outcomes and potential changes in health-related quality of life. The number of study visits was limited to 0, 12, and 24 months to minimize the burden on the healthcare system and participants, aligning with the real-world, observational design of the research. It is also important to note that improvements in quality-of-life responses may have been influenced by changing pandemic conditions, as baseline assessments were taken during the height of the COVID-19 pandemic, while final observations were collected in the post-pandemic period. This external variability related to restrictions and lockdowns could have impacted participants’ responses, potentially affecting the interpretation of the quality-of-life results. Furthermore, the selection of the EQ-5D-3L questionnaire, despite its practical benefits, may not capture as much detail as the EQ-5D-5L. Lastly, cost–utility analyses rely on certain assumptions and models, which may introduce uncertainty and limit generalizability to different healthcare systems or contexts.

A key strength of this study is its real-world setting, providing valuable and practical insights into the cost–utility and quality of life outcomes of PCSK9 inhibitors. This study is the first of its kind to evaluate these factors in a real-world context, thus supporting informed decision-making in clinical practice.

## 6. Material and Methods

### 6.1. Study Design

The MEMOGAL STUDY (NCT04319081) [20] is a multicenter, prospective study conducted in 12 Spanish hospitals with a double-arm, phase IV, open-label design. This study enrolled patients with FH or established CVD who were initiating PCSK9 inhibitors treatment for the first time with a follow-up of 2 years. The protocol received approval from the ethics committee and the Spanish Agency for Medicines and Health Products. Figure 1 illustrates the study design and patient disposition. In accordance with the Consolidated Health Economic Evaluation Reporting Standards (CHEERS) guidelines, this study has been conducted and reported to ensure comprehensive and transparent reporting of health economic evaluations.

### 6.2. Population

The inclusion criteria for this sub-analysis involved individuals over 18 years old who received their first prescription for a PCSK9 inhibitor: evolocumab (140 mg every 2 weeks) or alirocumab (75 mg or 150 mg every 2 weeks). Eligible participants had established atherosclerotic cardiovascular disease or hypercholesterolemia. Established atherosclerotic cardiovascular disease was defined as having a history of myocardial infarction, stroke, or peripheral arterial disease, while hypercholesterolemia encompassed homozygous familial hypercholesterolemia, heterozygous familial and non-familial hypercholesterolemia, or mixed dyslipidemia.

A total of 158 subjects met the inclusion criteria for this sub-analysis and provided written informed consent. This study was conducted in compliance with the principles of the Declaration of Helsinki.

### 6.3. Study Procedures

The inclusion period for this study spanned from May 2020 to May 2022, while the follow-up period extended from May 2020 to February 2023. The EQ-5D-3L questionnaire was administered at baseline, 12 months, 24 months, or during the final visit for patients with a follow-up duration exceeding or falling short of 24 months. The responsibility for questionnaire administration during study visits rested with the investigators, as listed in Appendix A. The investigators received instructions at the start of the study and during several investigator meetings throughout the follow-up period.

### 6.4. End Points

The primary objective was to conduct a cost–utility analysis over a 2-year period, utilizing the QoL and the associated costs of PCSK9 inhibitors compared to basic therapies such as statins or ezetimibe. In this study, the population included for comparison purposes was drawn from the same patient cohort and collected retrospectively, as we assessed the quality of life and costs before the initiation of PCSK9 inhibitor treatment and after two years of treatment. Furthermore, cost events and the costs of premature deaths were calculated. The MACEs analyzed were as follows: myocardial infarction, unstable angina, percutaneous coronary intervention, cardiac surgery (bypass), cardiovascular death, and death for any cause.

The secondary objective included evaluating changes in quality of life using the EQ-5D-3L questionnaire. This questionnaire consists of two sections: the EQ-5D-3L descriptive system, which assesses 5 dimensions (MOBILITY, SELF-CARE, USUAL ACTIVITIES, PAIN/DISCOMFORT, and ANXIETY/DEPRESSION) at three levels (no problems, some problems, extreme problems); and the EQ-5D-3L visual analogue scale (VAS) ranging from 0 to 100 points [19]. Assessments were conducted at baseline, 12 months, 24 months, and/or at the end of the study. A sample of the EQ-5D-3L is provided in Appendix A. 

### 6.5. Cost–Utility Analysis

The cost–utility analysis aimed to assess the economic value of the intervention, primarily using QoL as the outcome measure. Utility values were derived from the EuroQol Five-Dimensional Questionnaire (EQ-5D-3L), administered at baseline and follow-up assessments. The EQ-5D-3L provides values for all health states, resulting in an index for each of the 243 possible health states, including death, reflecting population preferences [21]. Direct healthcare costs, including medication expenses and healthcare resource utilization, were collected from medical records and billing databases [22]. Indirect costs, such as productivity loss, were estimated using standard approaches. Cost–effectiveness ratios, expressed as the incremental cost per QoL gained, were calculated to evaluate the efficiency of the intervention compared to alternative treatment strategies.

Regarding the cost perspective, the analysis adopts a societal viewpoint, encompassing both direct healthcare costs and indirect societal costs. Direct healthcare costs were collected from medical records and billing databases. Indirect costs, such as productivity loss, were estimated using standard approaches. Furthermore, discounting was applied to both costs and QoL at a rate of 3.5%, in line with the standard practice in economic evaluations in healthcare [23].

To provide a comprehensive overview of the Spanish healthcare system, we should mention that this study was conducted within the framework of the Spanish National Health System, which provides universal coverage to the Spanish population.

In the case of premature death costs, two distinct scenarios were considered:

For the active population: the expected benefits in terms of reduced incidence, mortality, and potential years of lost work-life were estimated based on the economic value derived from lost wages based on the average gross income per worker in the area (EUR 20,286 per year).

For the working-age population, the cost was estimated taking into account the average gross income of the worker (EUR 18,768.21 per year) and the unemployment rate (9.37%) in our healthcare area as of 30 April 2023 [24]. On the other hand, leisure time was valued at 47% of the cost of working hours [25].

For the retired population: The expected benefits in terms of reduced incidence, mortality, and potential years of life lost (relative to the average life expectancy) were estimated, considering the economic value derived from the contribution of individuals aged 65 and older to volunteer work and grandchild care (IPREM: EUR 600 per month). For the retired population, the cost was estimated taking into account the percentage of individuals aged 65 or older engaged in volunteer work, according to the CIS-IMSERSO study (2.3%) [26]; and those who dedicate themselves to grandchild care according to the study “Living Conditions of Older People” conducted by the Center for Sociological Research (22.6%) (Center for Sociological Research. Study “Living Conditions of Older People”) [27].

### 6.6. Statistical Analysis

For the primary and secondary endpoints related to EQ-5D-3L domains, statistical analysis was performed using the Fisher test. Also, frequencies and percentages were calculated to describe these domains. To test de VAS scale, means with their 95% confidence intervals (CIs) at and the standard deviation were calculated. Differences were assessed by applying Student’s t test to paired data. Both for the EQ-5D-3L and for the VAS test, data at the baseline time were compared with data at the end of follow-up.

Statistical difference was accepted at *p* < 0.05. All analyses were performed using SPSS 19.0 (IBM Corp. Released 2010. IBM SPSS Statistics for Windows, Version 19.0. IBM Corp., Armonk, NY, USA)

## 7. Conclusions

The current study represents a significant contribution to the literature by being one of the first real-world cost–utility analyses conducted on the use of PCSK9 inhibitors. Our results not only demonstrate favorable cost–utility outcomes but also reveal an improvement in all domains of the EQ-5D-3L questionnaire. This suggests that the incorporation of PCSK9 inhibitors into routine clinical practice is not only economically justified but also holds the potential to enhance patients’ quality of life. These findings provide valuable insights for healthcare decision-makers along with reimbursement policies, as they highlight the potential benefits of PCSK9 inhibitors in terms of both cost and utility.

## Figures and Tables

**Figure 1 diseases-12-00244-f001:**
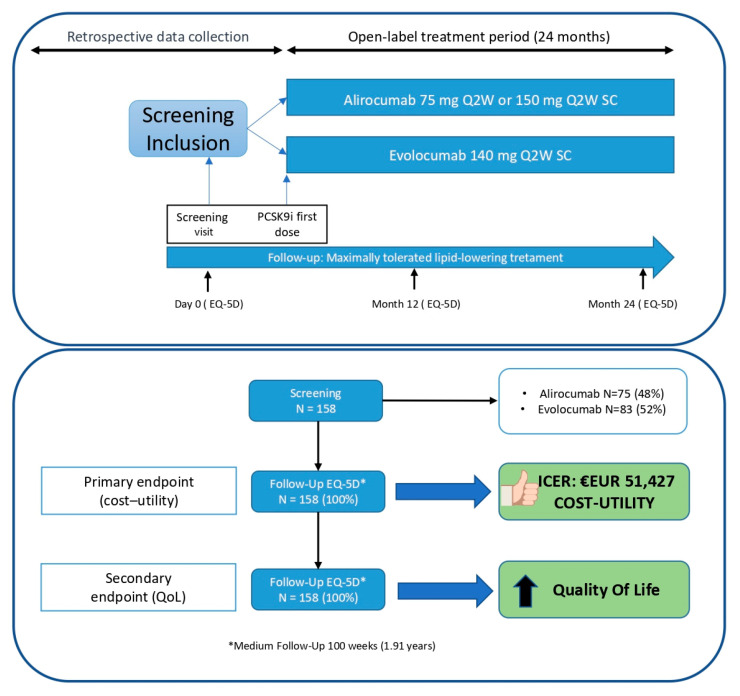
Study design and patient disposition.

**Figure 2 diseases-12-00244-f002:**
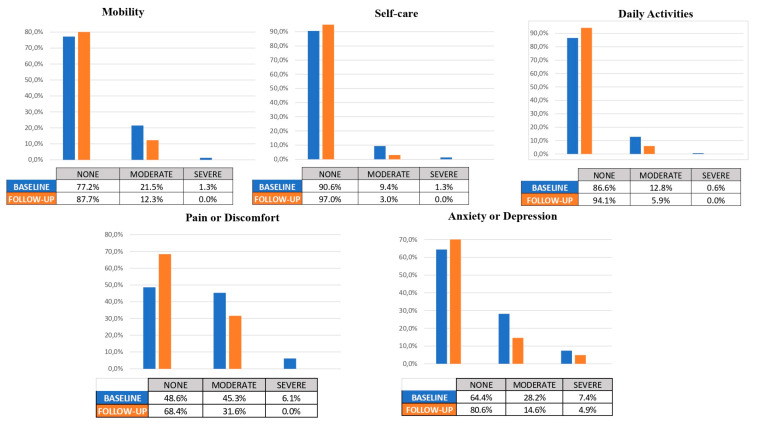
Secondary endpoint: changes in QoL.

**Table 1 diseases-12-00244-t001:** Demographic, baseline characteristics and treatments. This table provides a detailed summary of the demographic characteristics and treatment history of the study population, including age, sex, medical history, and specific treatments received (e.g., types of PCSK9 inhibitors and statins). The data include percentages and mean values, highlighting key baseline metrics like LDL cholesterol levels.

Demographic, Baseline Characteristics and Treatments	
Sex (male); n (%)	105 (66.50)
Years; mean (SD)	60.6 (10.20)
Height; mean (SD)	1.67 (0.08)
Weight; mean (SD)	81.0 (15.70)
Medical history; n (%)	
Cardiovascular disease	134 (84.80)
Familiar hypercholesterolemia	39 (24.70)
Statins intolerance	69 (43.70)
Dementia history	31 (19.60)
Diabetes	35 (22.20)
Hypertension	87 (55.10)
Heart failure	27 (17.10)
Diet	114 (72.20)
Smoking status; n (%)	
Current	17 (10.80)
Past smoker	85 (53.80)
Never	56 (35.40)
PCSK9 inhibitors; n (%)	
Alirocumab 150 mg	65 (41.10)
Alirocumab 300 mg	18 (11.40)
Evolocumab 240 mg	75 (47.50)
Statins; (%)	
Rosuvastatin	33.50
Atorvastatin	18.40
Pitavastatin	3.20
Other statins	1.20
Ezetimibe	58.60
LDL-c; mg/dL (SD)	
Baseline	145.18 (43,43)
Follow-up	62.11 (57.00)

**Table 2 diseases-12-00244-t002:** Events and costs. This table lists the clinical events (e.g., myocardial infarction, unstable angina, stroke, PCI, CABG, and cardiovascular death) observed during the study and the associated costs for both the PCSK9 inhibitor and standard therapy groups. It includes the Diagnosis-Related Group (DRG) codes and costs in EUR, providing a detailed economic evaluation of the treatments.

	IAM_No.	Angina Inestable_No.	Stroke_No.	PCI_No.	CABG_No.	Death_cv
**PCSK9i**	3	2	0	1	0	2
**Standard**	4	3	1	2	2	0
Death due to other causes were not included into analysis in both arms (2 deaths due to cancer).

		**DRG**	**COST (EUR )**
		Myocardial Infarction	5183.35
		Unstable Angina	4594.27
		Stroke	3748.38
		Percutaneous Coronary Intervention	10843.58
		Cardiac Surgery (Bypass)	29500.06
Ministry of Health. Average costs of Diagnosis-Related Groups (DRG). Available at: https://www.sanidad.gob.es/estadEstudios/estadisticas/inforRecopilaciones/anaDesarrolloGDR.htm (accessed on 7 August 2024).

**Table 3 diseases-12-00244-t003:** Results of pharmacological treatment costs and quality of life. This table compares the costs of pharmacological treatments (PCSK9 inhibitors vs. standard therapy) and the associated health-related quality of life (QoL) outcomes over two years. It details the incremental cost–effectiveness ratio (ICER) calculation, demonstrating the cost–utility of PCSK9 inhibitors compared to standard therapies.

Results	Amount (EUR )
Pharmacological treatment cost PCSK9 inhibitors	EUR 13,633.39
Pharmacological treatment cost standard therapy	EUR 3638.25
Total event cost PCSK9 inhibitors	EUR 35,582.17
Unit event cost PCSK9 inhibitors	EUR 223.78
Total event cost standard therapy	EUR 118,951.87
Unit event cost standard therapy	EUR 748.12
Total premature death cost PCSK9 inhibitors	EUR 80,843.68147
Total premature death cost standard therapy	EUR 0
Unit cost of events + mortality PCSK9 inhibitors	EUR 732.23
Unit cost of events + mortality standard therapy	EUR 748.12
Discount rate: 3.5%	
Total unit cost for pharmacological treatment PCSK9 inhibitors after discount rate	EUR 13,410.46
Total unit cost for pharmacological treatment standard therapy after discount rate	EUR 4094.72
QoL PCSK9 inhibitors (2 years)	1.648851948
QoL PCSK9 inhibitors (2 years) after discount rate	1.53922094
QoL standard therapy (2 years)	1.454807792
QoL standard therapy (2 years) after discount rate	1.35807864
ICER: (Total unit cost PCSK9 inhibitors—Total unit cost standard therapy)/(QoL PCSK9 inhibitors—QoL standard therapy)	EUR 51,427.72

**Table 4 diseases-12-00244-t004:** Secondary endpoint—changes in quality of life. This table shows the distribution of patients reporting levels of problems (none, moderate, severe) within the EQ-5D-3L dimensions at baseline and follow-up. It includes *p*-values indicating the statistical significance of changes observed in each domain, highlighting improvements in mobility, self-care, daily activities, pain/discomfort, and anxiety/depression.

	MOBILITY		SELF-CARE		DAILY ACTIVITIES		PAIN/ DISCOMFORT		ANXIETY/ DEPRESSION	
	Baseline	FU	Baseline	FU	Baseline	FU	Baseline	FU	Baseline	FU
None	77.20	87.70	90.60	970	86.60	94.10	48.60	68.40	64.40	80.60
Moderate	21.50	12.30	9.40	3.00	12.80	5.90	45.30	31.60	28.20	14.60
Severe	1.30	0.00	1.30	0.00	0.00	0.00	6.10	0.00	7.40	4.90
*p*-value	<0.001		<0.001		<0.001		<0.001		<0.001	

	**EVA**		**Baseline**		67.04( ± 20.069)		66.62–72.67		0.086	
			**End point**		69.64( ± 18.683)		63.81–70.27			

**Table 5 diseases-12-00244-t005:** Spanish population norms (EQ-5D-3L). This table compares the baseline and follow-up quality of life data for the study population with the general Spanish population norms. It provides proportions of respondents reporting problems in various dimensions and includes self-reported EQ VAS ratings by age group and gender, showing how the study population’s QoL measures up against national averages.

	MOBILITY		SELF-CARE		DAILY ACTIVITIES		PAIN/ DISCOMFORT		ANXIETY/ DEPRESSION	
	Females	Males	Females	Males	Females	Males	Females	Males	Females	Males
None	84	89	94	97	86	92	73	23	90	95
Problems	16	11	6	3	14	8	27	17	10	5

			**EVA**				**Females**		71	
							**Males**		73	

Janssen MF, Szende A, Cabases J, Ramos-Goñi JM, Vilagut G, König HH. Population norms for the EQ-5D-3L: A cross-country analysis of population surveys for 20 countries. [9]

## Data Availability

The data supporting the reported results can be shared upon request. The data and materials used in this research will be available on demand for readers interested in replicating or building upon our study. Interested parties can contact the corresponding author to request access to the data and materials. We will ensure that the requested information and materials are provided in a timely and complete manner.

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
