# Peer review of "Cost–Utility Analysis of PCSK9 Inhibitors and Quality of Life: A Two-Year Multicenter Non-Randomized Study"

_diseases, 2024, doi:10.3390/diseases12100244_

Round 1

Reviewer 1 Report

Comments and Suggestions for Authors

- the role of novel siRNA in lipid management should be discussed. Authors can consider the paper from Scicchitano P, et al. Biomed Pharmacother. 2021 Nov;143:112227. 

Author Response

The role of novel siRNA in lipid management should be discussed. Authors can consider the paper from Scicchitano P, et al. Biomed Pharmacother. 2021 Nov;143:112227.

We appreciate the reviewer’s suggestion, and we have incorporated the article by Scicchitano et al. into the discussion section. This article provides valuable information regarding the use of siRNA, demonstrating an approximate 50% reduction in LDL-C plasma concentrations. We have added this reference in the context of LDL reduction alongside other lipid-lowering treatments such as PCSK9 inhibitors to offer a more comprehensive perspective on the available therapies for lowering LDL-C levels.

Line 234 to 237 (reference 27).

Reviewer 2 Report

Comments and Suggestions for Authors

For the cost-utility analysis, the authors use the treatment group itself as a control by considering retrospective data before the initiation of iPCSK9 when the group was receiving standard treatment. This approach, while convenient given the data available, has some weaknesses that should be discussed in more detail. First, the treatment group by construction is older, and may have experienced deterioration in health status generally as well as owing to the presumably less effective standard course of treatment. Second, the enrolment period corresponds with the COVID-19 pandemic while the ‘control’ period predates it. This might have affected health status and health outcomes for the treatment group. Third, it appears that those individuals who suffered major health events during the control period remain in the treatment sample, though they would presumably have received significant intervention (such as cardiac bypass surgery) that could change their likelihood of such an event occurring again in the treatment period.

The calculation of costs associated with PCSK9i are very significantly affected by two deaths in the treatment period (cost 80,843) versus no deaths in the control period, and the inclusion of these costs brings the unit cost of events+mortality much closer to equal between the treatment and control approaches. More generally, given the very low actual counts of mortality and major adverse health events, some random sample variation could have changed the calculated benefits substantially. Is the estimated difference statistically significantly different from zero?

Although the authors do not have estimates of changes in quality of life for a control group, they benchmark their estimates against population averages. Could they instead use as a control estimate a subset of individuals of the same age range and location of residence rather than the overall population?

Significant improvements in quality of life measures were observed over the study period, but I note that the baseline responses were provided at the height of the pandemic, with its related lock-downs and restrictions. Observations at the end of the study period were taken in the post-pandemic period. Such changing conditions might have contributed to the improvement in quality of life responses in ways unrelated to the iPCSK9 treatment.

Author Response

For the cost-utility analysis, the authors use the treatment group itself as a control by considering retrospective data before the initiation of iPCSK9 when the group was receiving standard treatment. This approach, while convenient given the data available, has some weaknesses that should be discussed in more detail. First, the treatment group by construction is older, and may have experienced deterioration in health status generally as well as owing to the presumably less effective standard course of treatment. Second, the enrolment period corresponds with the COVID-19 pandemic while the ‘control’ period predates it. This might have affected health status and health outcomes for the treatment group. Third, it appears that those individuals who suffered major health events during the control period remain in the treatment sample, though they would presumably have received significant intervention (such as cardiac bypass surgery) that could change their likelihood of such an event occurring again in the treatment period.

The calculation of costs associated with PCSK9i are very significantly affected by two deaths in the treatment period (cost 80,843) versus no deaths in the control period, and the inclusion of these costs brings the unit cost of events+mortality much closer to equal between the treatment and control approaches. More generally, given the very low actual counts of mortality and major adverse health events, some random sample variation could have changed the calculated benefits substantially. Is the estimated difference statistically significantly different from zero?

The reviewer is absolutely right in suggesting a potential bias in this respect. To address this, we performed a paired Student’s t-test to evaluate whether the difference in mortality between the pre- and post-PCSK9i treatment periods is statistically significant. The results show that the differences in mortality are random and not dependent on the therapy followed, but rather influenced by other factors.

It is also worth noting that, without the two deaths (which may be due to random variation), the benefits of the PCSK9i programme would be higher. Therefore, even in this negative scenario for the PCSK9i comparison, the programme remains more cost-effective, leading the authors to conclude that the results are robust in favor of PCSK9i.

death_post-PCSK9i

death_pre-PCSK9i

Mean

0,01290323

0,01290323

Variance

0,01281944

0,01281944

Observations

155

155

Pearson's correlation coefficient

-0,0130719

Hypothetical difference of means

0

Degrees of freedom

154

t-statistic

0

P(T<=t) one-tailed

0,5

Critical value of t (one-tailed)

1,65480839

P(T<=t) two-tailed

1

Critical value of t (two-tailed)

1,97548806

Student's t-test for paired samples shows that the differences in mortality are random.

Although the authors do not have estimates of changes in quality of life for a control group, they benchmark their estimates against population averages. Could they instead use as a control estimate a subset of individuals of the same age range and location of residence rather than the overall population?

The reviewer's suggestion is correct, but it should also be noted that a standardised mortality ratio (SMR) is interpretable in terms of relative risk (RR); the RR of the model refers to the estimated effects in the trial population, while the SMR represents the overall effect for the population as a whole. (Gorini G, Zappa M, Miccinesi G, Paci E, Seniori Costantini A. Breast cancer mortality trends in two areas of the province of Florence, Italy, where screening programmes started in the 1970s and 1990s. Br J Cancer. 2004; 90:1780-3).

Significant improvements in quality of life measures were observed over the study period, but I note that the baseline responses were provided at the height of the pandemic, with its related lock-downs and restrictions. Observations at the end of the study period were taken in the post-pandemic period. Such changing conditions might have contributed to the improvement in quality of life responses in ways unrelated to the iPCSK9 treatment.

We appreciate the reviewer’s valuable comments, which undoubtedly enrich the discussion of our study. We agree that the changing conditions between the onset of the pandemic and the post-pandemic period may have influenced the quality of life responses. As the reviewer suggests, this is a factor that could have contributed to the improvements observed, independent of the iPCSK9 treatment.

To address this concern, we have incorporated additional discussion in the limitations section of the study, acknowledging that the COVID-19 pandemic and the associated restrictions during the study period may have affected the baseline responses of the participants. This potential external bias will be mentioned as a possible source of variability that could influence the interpretation of the quality of life results. Despite these external factors, we believe the positive impact of iPCSK9 treatment remains significant, although we recognize that this aspect warrants further investigation.

Lines 333 to 338.

Reviewer 3 Report

Comments and Suggestions for Authors
  1. The study involved only 158 patients across 12 hospitals in Spain, which may limit the generalizability of the findings. A larger and more diverse sample could provide a more comprehensive understanding of the cost-effectiveness of PCSK9 inhibitors across different populations and healthcare settings.
  2.  Conducted over two years, the follow-up period may not be sufficient to capture the long-term benefits or adverse effects of PCSK9 inhibitors. Longitudinal studies are needed to assess the sustainability of health-related quality of life (QoL) improvements and the economic impact over a more extended period.
  3. The study utilized the EQ-5D-3L questionnaire to assess QoL. While this tool is widely used, it may not capture all dimensions of health relevant to patients with cardiovascular disease. A more comprehensive assessment tool could provide deeper insights into the specific impacts of PCSK9 inhibitors on patients' lives.

Comments on the Quality of English Language

English edits required

Author Response

The study involved only 158 patients across 12 hospitals in Spain, which may limit the generalizability of the findings. A larger and more diverse sample could provide a more comprehensive understanding of the cost-effectiveness of PCSK9 inhibitors across different populations and healthcare settings.

Conducted over two years, the follow-up period may not be sufficient to capture the long-term benefits or adverse effects of PCSK9 inhibitors. Longitudinal studies are needed to assess the sustainability of health-related quality of life (QoL) improvements and the economic impact over a more extended period.

First of all, I would like to thank the reviewer for their valuable recommendations. For this new revision, an expert in the translation of scientific literature from our department has reviewed the English of the manuscript, ensuring greater precision in the writing.

Regarding the specific recommendations, we understand the concerns related to the sample size and follow-up duration. The sample of 158 patients accurately reflects the reality at that time, as the study was approved in January 2020, just before the onset of the COVID-19 pandemic. Patient inclusion occurred during a period when the number of medical consultations and prescriptions was significantly reduced due to pandemic restrictions. Despite this, we believe that the sample is representative of the clinical context during that period, capturing a particular but relevant situation in clinical practice.

As for the two-year follow-up period, since this is a real-world, non-commercial study, both patient retention and the number of visits were limited to standard clinical practice. Additionally, the fact that this is an independent study, without the involvement of a commercial sponsor, also influenced the study’s design and scope, limiting the possibility of adding more visits or extending the follow-up period. Nevertheless, we agree that longer-term longitudinal studies would be valuable to assess the long-term benefits and potential adverse effects of PCSK9 inhibitors.

The study utilized the EQ-5D-3L questionnaire to assess QoL. While this tool is widely used, it may not capture all dimensions of health relevant to patients with cardiovascular disease. A more comprehensive assessment tool could provide deeper insights into the specific impacts of PCSK9 inhibitors on patients' lives.

The reviewer's assessment is correct in that two questionnaires, one more generic and one more specific, would be complementary and provide more information. However, the difficulty created because the ex ante control group received the EQ5D questionnaire for bureaucratic reasons and no specific questionnaire limits the possibility of comparing questionnaires. Furthermore, there is literature that considers the use of a generic questionnaire such as the EQ5D (which also has the advantage of simplicity from the point of view of its management) to be more than acceptable from a methodological point of view.

(O.J. Wouters, H. Naci, N.J. Samani. QALYs in cost-effectiveness analysis: an overview for cardiologists. Heart., 101 (2015), pp. 1868-1873

http://dx.doi.org/10.1136/heartjnl-2015-308255 | Medline

  1. Barradas-Pires, A. Constantine, K. Dimopoulos. Percutaneous Interventions in Adult Congenital Heart Disease. Patient Reported Outcomes and Quality of Life in Cardiovascular Interventions, pp. 171-184

J.L. Pinto Padres, F.I. Sánchez Martínez, J.M. Abellán Perpiñán. Métodos Para La Evaluación Económica de Nuevas Prestaciones. Centre de Recerca en Economia i Salut – CRES Ministerio de Sanidad y Consumo, (2003)

Davidson, T., et al., Cost-effectiveness of dabigatran compared with warfarin for patients with atrial fibrillation in Sweden. European Heart Journal, 2013. 34(3): p. 177-183

Wouters, Olivier J., Naci, Huseyin and Samani, Nilesh J (2015) QALYs in cost-effectiveness analysis: an overview for cardiologists. Heart. ISSN 1355-6037)

Reviewer 4 Report

Comments and Suggestions for Authors

Dear Authors,

            I have read with great interest your original study on the cost utility of PCSK9 inhibitors. The paper is well-written, interesting, and brings novelty to the field. However, there are several issues that need to be addressed:

          -  Quality of life and MACE are influenced by any treatment targeting the underlying cardiovascular condition of the patient, not solely by dyslipidemia medication. In this regard, it would be valuable to provide information about concomitant medication, especially whether there were any changes in medication during the follow-up period, given that the control group and the study group are the same set of patients. If any changes in concomitant medication occurred, they should be acknowledged in the limitations section.

          -  Ensure that abbreviations are defined the first time they appear and only once. For example, "iPCSK9" is first defined in the materials and methods section, which, in the current format, appears at the end of the paper.

Author Response

Dear Authors,

            I have read with great interest your original study on the cost utility of PCSK9 inhibitors. The paper is well-written, interesting, and brings novelty to the field. However, there are several issues that need to be addressed:

          -  Quality of life and MACE are influenced by any treatment targeting the underlying cardiovascular condition of the patient, not solely by dyslipidemia medication. In this regard, it would be valuable to provide information about concomitant medication, especially whether there were any changes in medication during the follow-up period, given that the control group and the study group are the same set of patients. If any changes in concomitant medication occurred, they should be acknowledged in the limitations section.

First of all, we would like to thank the reviewer for their insightful comments on our article. Regarding the first point, we find this observation particularly interesting, as it is true that concomitant medication could have a significant impact on the outcomes.

We would like to clarify that the patients in this study were followed in the pharmacy and cardiology clinics outpatients of the same country, adhering to current clinical practice guidelines that are in line with European recommendations (ESC) for cardiovascular risk factor control. Specifically, in terms of LDL-c control, these patients received PCSK9 inhibitors, which are reimbursed for those with LDL-c levels >100 mg/dL despite being on rosuvastatin 20 mg + ezetimibe 10 mg or atorvastatin 80 mg + ezetimibe 10 mg, having already tried at least two maximum dose statins. Alternatively, another reason for reimbursement is total or partial intolerance to statins. This ensures a stable lipid-lowering treatment throughout the follow-up period, very similar to what occurs in clinical trials.

          -  Ensure that abbreviations are defined the first time they appear and only once. For example, "iPCSK9" is first defined in the materials and methods section, which, in the current format, appears at the end of the paper.

Thank you once again for your valuable comments. In response to your latest suggestion, we have carefully corrected all abbreviations to ensure they are properly defined the first time they appear and used consistently throughout the manuscript. These changes have been highlighted using the track changes function for your review.

Round 2

Reviewer 2 Report

Comments and Suggestions for Authors

no additional comments

Reviewer 3 Report

Comments and Suggestions for Authors

Manuscript can be accepted in current form

Reviewer 4 Report

Comments and Suggestions for Authors

Dear Authors,

Thank you for addressing the issues raised.